



# Leaf wax *n*-alkane distributions record ecological changes during the Younger Dryas at Trzechowskie paleolake (Northern Poland) without temporal delay

Bernhard Aichner[1], Florian Ott[2,3], Michał Słowiński[2,4], Agnieszka M. Noryśkiewicz[5], Achim Brauer[2],
Dirk Sachse[6]

[1]Inst. of Earth and Environmental Sciences, University of Potsdam, 14476 Potsdam-Golm, Germany
[2]Section 5.2: Climate Dynamics and Landscape Evolution, GFZ German Research Centre for Geosciences, 14473 Potsdam, Germany
[3]Department of Archaeology, Max Planck Institute for the Science of Human History, 07745 Jena, Germany
[4]Department of Environmental Resources and Geohazards, Institute of Geography and Spatial Organisation, Polish Academy of Sciences, 00-818 Warsaw, Poland
[5]Institute of Archaeology, Nicolaus Copernicus University, 87-100 Toruń, Poland
[6]Organic Surface Geochemistry Lab, Section 5.1: Geomorphology, GFZ German Research Centre for Geosciences, 14473 Potsdam, Germany

*Correspondence to*: Bernhard Aichner (bernhard.aichner@gmx.de)

**Abstract.** While of higher plant origin, a specific plant source assignment of sedimentary leaf wax *n*-alkanes remains difficult. Recent compilations of global plant data sets have demonstrated an overlapping and non-systematic production of different chain-length homologues among different classes of terrestrial vegetation. Further, *n*-alkane distributions can change within the same species due to environmental changes. In addition, it is unknown how fast a changing catchment vegetation would be reflected in sedimentary leaf wax archives. However, in particular for a quantitative interpretation of *n*-alkane C and H isotope ratios in terms of paleohydrological and paleoecological changes, a better understanding of transfer times and dominant sedimentary sources of leaf wax *n*-alkanes is required.

In this study we aim to identify the major leaf wax contributors to a Central European lacustrine system. Therefore, we tested to what extent leaf wax *n*-alkane compositional changes (expressed through compound concentration ratios, such as $nC_{27}$ vs. $nC_{31}$, average chain length ACL, etc.) can be linked to known vegetation changes, specifically during the Younger Dryas cold period (YD), by comparison with high-resolution palynological data from the same archive. We analysed leaf wax *n*-alkane concentrations and distributions in decadal resolution from a sedimentary record from Trzechowskie paleolake –TRZ – (Northern Poland), covering the Late Glacial to early Holocene (13,360 – 9,940 yrs BP). As additional source indicator of targeted *n*-alkanes, compound specific carbon isotopic data have been generated in lower time resolution.

The results showed rapid responses of *n*-alkane distribution patterns coinciding with major climatic and paleoecological transitions. We find a shift towards higher ACL values at the Allerød/YD transition between 12,680 and 12,600 yrs BP, coeval with a decreasing contribution of arboreal pollen (mainly *Pinus* and *Betula*) and a subsequently higher abundance of



pollen derived from herbaceous plants (Poaceae, Cyperaceae, *Artemisia*), as well as shrubs and dwarf shrubs *Juniperus* and *Salix*. The termination of the YD was characterized by a successive increase of *n*-alkane concentrations coinciding with a sharp decrease of ACL values between 11,580 – 11,490 years BP, reflecting the expansion of woodland vegetation at the YD/Holocene transition. Centennial reversals to longer chain lengths during the Allerød could possibly be linked to

Greenland Interstadial 1b (GI-1b). A similar pattern during the early Holocene has more likely been triggered by rapid ecological responses in course of warming, rather than to reflect a local impact of a Preboreal Oscillation or 11.4 yr event. Another gradual increase in ACL values after 11,200 yrs BP, together with decreasing *n*-alkane concentrations, most likely reflects the early Holocene vegetation succession with a decline of *Betula*. These results show, that *n*-alkane distributions reflect vegetation changes and that a fast (i.e. subdecadal) signal transfer occurred. However, our results also indicate that a

standard interpretation of directional changes in biomarker ratios remains difficult. Instead, responses such as changes of ACL need to be discussed in context of other proxy data. In addition, we find that organic geochemical data integrate different ecological information compared to pollen, since some gymnosperm species, such as *Pinus*, produce only very low amount of *n*-alkanes and thus their contribution may be largely absent from biomarker records. Our results demonstrate that a combination of palynological and *n*-alkane data can be used to infer the major sedimentary leaf wax sources and constrain

leaf wax transport times from the plant source to the sedimentary sink and thus pave the way towards quantitative interpretation of compound specific hydrogen isotope ratios for paleohydrological reconstructions.

## 1. Introduction

In the past decades organic geochemical proxies, such as concentrations and -ratios of *n*-alkanes derived from aquatic and terrestrial organisms, have increasingly been used as paleoecological indicators. These compounds have been postulated to

be at least semi-source-specific for different groups of organisms e.g. aquatic vs. terrestrial plants, trees vs. grasses, gymnosperms vs. angiosperms (e.g. Eglinton and Hamilton, 1967; Wakeham, 1976; Meyers and Ishiwatari, 1993; Ficken et al., 2000). Consequently, changes of *n*-alkane ratios in sedimentary records have frequently been interpreted to reflect changes of vegetation in lakes and their catchment (e.g. Schwark et al., 2002; Hanisch et al., 2003; Liu and Huang, 2005; Zhang et al., 2006; Nichols et al., 2006; Hockun et al., 2016). While most of these studies delivered plausible indications that

application of these proxies reliably reflect changes of local ecological conditions, a compilation of published plant leaf wax *n*-alkane data revealed that there is little agreement between specific *n*-alkanes and potential groups of source organisms at least if considering a global data set (Bush and McInerney, 2013). Another important aspect to consider is variability of leaf wax concentrations, which appears to be high among different plants species (Diefendorf et al., 2011). This suggests that sedimentary *n*-alkanes rather reflect a signal biased towards strong leaf wax producers within the catchment instead of the

average composition of vegetation. Finally, transit times from source to sink need to be considered, since there is evidence for significant ageing of leaf wax compounds e.g. in soils from some lacustrine archives, leading to potential lag times before being deposited in sediments (Douglas et al., 2014).



Here we focus on the transition from the Late Glacial to the Holocene, which was characterized by abrupt climatic and vegetation changes. After an initial warming phase (Bølling/Allerød) temperatures in western Europe rapidly dropped by ca. 4-6°C for ca. 1100 years, possibly caused by a reduction of the Northern Atlantic meridional overturning circulation (AMOC), forced by meltwater input into the North Atlantic (Rasmussen et al., 2006; McManus et al., 2004; Denton et al.,
2010; Elmore et al., 2011; Heiri et al., 2014; Renssen et al, 2015). This Younger Dryas cold period (YD) was originally defined by means of ecological changes in Europe inferred from lacustrine palynological records (Firbas 1949, 1954; Iversen, 1973; Overbeck, 1975). Attempts to precisely date the onset and termination of this event included the establishment of chronologies based on annually laminated (varved) sediments which are anchored by tephra layers, such as the Laacher See Tephra (LST), Vedde Ash, Askja-S, or Hässeldalen (Brauer et al., 1999a,b; Goslar et al., 2000; Neugebauer et al., 2012;
Lane et al., 2013; Wulf et al., 2013, 2016; Ott et al., 2016). For instance, applying these techniques, at Meerfelder Maar (MFM) in western Germany the YD was dated to span from 12,680 – 11,590 varve years BP (Brauer et al., 1999a,b). These studies have shown that vegetation changes in continental Europe at the YD-onset were not synchronous to the cooling in Greenland (i.e. the onset of Greenland Stadial 1; GS-1) but occurred with a lag of ca. 100-170 yrs (Brauer et al., 1999b; Lane et al., 2013; Rach et al., 2014). By reconstructing hydrological conditions in western Germany using compound-specific
hydrogen isotope analysis of biomarkers in subdecadal resolution, Rach et al. (2014) suggested that aridification occurring ca. 170 yrs after the onset of the cooling in Greenland was the main trigger for the vegetation response. As a potential cause, they suggest expansion of sea-ice and a consequent southward migration of the Westerlies which then transported cooler and drier air to Europe (Brauer et al., 2008). Recently, Słowinski et al. (2017) found a further slight delay (20 yrs) of the ecological changes at the YD-onset in the more continental parts of eastern Europe compared to western Germany, based on
pollen data from Trzechowskie paleolake (TRZ), Northern Poland. Constraints about the termination of the YD are less precise. While Rach et al. (2014) observed a synchronous start of the Holocene in the MFM compared to Greenland ice cores, there is an ongoing debate about timing and nature of short-term cooling events, sometimes referred as Preboreal Oscillation (PBO) during the early Holocene (e.g. Björk et al., 1997, Bos et al., 2007; Ott et al., 2016).

In this study we focus on a ca. 3000 year record from Northern Poland, which spans from the Allerød to the early Holocene
and was sampled in ca. 10 yr resolution. The principal aim was to test to what extent leaf wax *n*-alkane compositional changes in lacustrine sediments can be linked to vegetation changes by comparison with published high-resolution palynological data from the same archive (Słowinski et al., 2017) which cover the Allerød to YD-transition. Another goal was to evaluate possible differences in the transport pathways of pollen vs leaf waxes, i.e. the identification of temporal leads or lags between leaf wax and pollen proxies. Further, we aim to establish a decadal, organic geochemical based
paleoecological record over the time-period of the YD.





## 2. Study site

The Trzechowskie paleolake (TRZ) is located in central Northern Poland ca. 75 km SSW of the city of Gdansk (TRZ; 53° 52.40 N, 18° 12.930 E; 110 m a.s.l.; Fig. 1) and developed after the retreat of the Late Weichselian ice sheet at ca. 15.8 cal. kyrs BP (Marks, 2012). It was formed by the successive melting of a buried ice block in a subglacial channel in the outwash

plain of the Wda river and initially formed one large lake with still existing Lake Czechowskie (Błaszkiewicz et al., 2015; Słowiński, 2010; Słowiński et al., 2015). The paleolakes maximum extension was approximately 1500 x 450m before it turned into peatland around 3,200 cal yrs BP (transition from gyttja to the peat at depth 183-182cm dated to 3,050 ± 40 BP). Finally, in the late 19th century most of the former wetlands in this area were drained with the purpose to gain agricultural grass land (Wulf et al., 2013; Słowinski et al., 2017). The present catchment is characterized by anthropogenic *Pinus*

*sylvestris* forests. While Spring and Fall are characterized by alternating winds, major wind directions are low speed westerlies during summers and easterlies during winters (Woś, 1999; Wulf et al., 2013). Mean annual precipitation and temperature are ca. 600 mm and 7 °C, respectively (1981-1998 period; Wójcik and Marciniak, 1993; Kozłowska-Szczesna, 1993).

## 3. Material and Methods

### 3.1. Sampling and age-model

A 137 cm long core section (1134 – 1271 cm composite depth; Fig. 1b), was sampled in 0.5 cm intervals, except for the uppermost 17 cm and lowermost 10 cm, which were sampled in 1 cm intervals. Within the laminated section (1244.5 – 1262 cm), the sampling strategy was adjusted to obtain a 10 years resolution (i.e. sampling of 10 varves), resulting in sample slices varying between 0.4 – 0.7 cm thicknesses.

The age-model was created with OxCal v4.2 (Bronk Ramsey, 2001; Bronk Ramsey, 2008; Bronk Ramsey, 2013) using a *P-sequence* model with variable *k parameter* and the most recent radiocarbon calibration dateset (IntCal13) (Reimer et al., 2013). The following age-anchor points have been used (Table 1): a) five [14]C AMS radiocarbon dates, calibrated with the Intcal13 data set (Reimer et al., 2013); b) two tephra layers (Askja-S: 1164cm, 11228 ± 226 yrs BP, age from Ott et al., 2016; LST: 1253.75 cm, 12880 ± 40 yrs BP, age from Brauer et al., 1999b), c) varve chronology within the laminated

section, anchored by the LST (Wulf et al., 2013); d) one biostratigraphic change, characterized by increase of *Pinus* together with the abrupt decrease of NAP and *Juniperus*. The latter has been synchronized with a similar shift in closeby Lake Gosciaz (1183.5 cm, 11515 ± 35 yrs BP) which is considered as the regional stratotype for the YD/Holocene boundary in Northern Poland (Ralska-Jasiewiczowa et al., 1992; Litt et al., 2001; Ott et al., 2016).

The age model of this study (Supplementary Figure 1) includes the Askja-S tephra and a biostratigraphic change (Ott et al.,

2016) as additional early Holocene anchor points, resulting in a slightly different age model than published earlier by Słowinski et al. (2017). Pollen and macrofossil data from this paper were transferred to the updated age-model. Since



Słowinski et al. (2017) focused on a core section which covers the Allerød/YD-transition (1230 – 1262cm), offsets between the old and updated ages within that part of the core were minor, i.e. within the range of 5 – 15 years and restricted to the non-laminated part at the YD onset (between 1230 and 1244.5 cm).

**3.2. Biomarker extraction and quantification**

After freeze-drying and grounding with a mortar, the sediments have been extracted for 2 cycles of 15 minutes with dichloromethane/methanol (9:1) using a Dionex ASE 350 accelerated solvent extraction system at 100°C and 75 bar. 5α-androstane was spiked as internal standard to the total lipid extract before the aliphatic fraction was eluted over silica-gel (0.040-0.063 mesh) with n-hexane as solvent. The n-alkanes were separated and quantified relative to an internal standard

using an Agilent 7890A gas chromatography (GC) instrument equipped with an Agilent 5975C Series mass selective detector (MSD) system and flame ionization detection (FID; Agilent 7683B Series). 1 μl of sample was injected onto a 30 m Restek DB-5 column (30 m, inner diameter 0.25 mm, film thickness 0.25 μm). Temperature was programmed from 70–320 °C at a rate of 12 °C/min (held 15 min).

**3.3. Compound-specific carbon isotope analysis**

Compound-specific carbon isotopic values were measured using gas chromatography isotope ratio mass spectrometry (GC-IRMS). We used a Thermo Scientific® Trace gas chromatograph equipped with a Restek RTX-5 column (30 m, inner diameter 0.25 mm, film thickness 0.25 μm) and a split-splitless (SSL) injector operated in splitless mode with an evaporation temperature of 60 °C. The GC was connected via a GC Isolink with combustion furnace (1000 °C) and a Conflo IV interface

to a DeltaVPlus isotope ratio mass spectrometer. Reference peaks of $CO_2$ bracket target compound peaks during the course of a GC-IRMS run. Two of these peaks were used for the standardization of the isotopic analysis, while the remainders were treated as unknowns to assess precision. Precision of these replicates was better than 0.6 ‰. Data were normalized to the Vienna Pee Dee Belemnite (VPBD) carbon isotopic scale by comparing them with an external standard containing 15 n-alkane compounds ($C_{16}$ to $C_{30}$ of known isotopic composition (A5-mix obtained from A. Schimmelmann, Indiana

University, Bloomington). The root mean square error of replicate measurements of the standard across the course of analyses was below 0.7%.

**3.4. Alkane parameters and statistics**

Alkane parameters have been calculated using the following equations:

Average chain length (ACL): $\Sigma(C_n \times n) / \Sigma(C_n)$; $Cn$: concentration of each n-alkane with $n$ carbon atoms.



Proxy for aquatic macrophytes ($P_{aq}$; Ficken et al., 2000): $P_{aq} = (nC_{23} + nC_{25}) / (nC_{23} + nC_{25} + nC_{29} + nC_{31})$

Before calculating Pearson Correlation Coefficients (PCCs) between $n$-alkane and pollen data, data have been z-transformed. Both datasets, consisting of 76 ($n$-alkanes) and 65 (pollen) data points, respectively, were transferred to equidistant time series with 65 data points, using the zoo-and simecol packages in R-Studio. PCCs were calculated in SPSS.

## 4. Results

Aliphatic lipids in the samples were dominated by odd-chain $C_{23}$, $C_{25}$, $C_{27}$, $C_{29}$, and $C_{31}$-$n$-alkanes, with $nC_{27}$ as the most abundant compound in all samples. The summed concentrations of these five compounds ranged between 3 to 263 µg / g dry weight (d.w.). Highest concentrations were measured before ca. 12,600 yrs BP and between ca. 11,300 – 10,400 yrs BP (Fig. 2). The top of the studied core interval is marked by a 4 cm thick organic rich layer (Fig. 1b), which was characterized by enhanced $n$-alkane concentrations but unchanged $n$-alkane patterns.

Throughout the record, we observed significant changes in the relative abundances of target compounds, resulting in pronounced shifts of related parameters such as ACL, $P_{aq}$, and $n$-alkane ratios $nC_{27}/(nC_{27} + nC_{29})$, $nC_{27}/(nC_{27} + nC_{31})$, $nC_{29}/(nC_{29} + nC_{31})$. For instance, ACL significantly increased between ca. 12,680 – 12,600 and decreased between 11,580 – 11,490 yrs BP. Minor oscillations of these indices were observed during the Allerød and early Holocene.

20 samples were analyzed for carbon isotope ratios (expressed as $\delta^{13}C$ values). These values were relatively constant (ca. -33‰) for the $nC_{31}$-alkane while $nC_{29}$ $\delta^{13}C$ values became slightly more positive (up to -31.5‰) throughout the studied interval (12,650 – 11,200 yrs BP). Other $n$-alkanes were characterized by increasing enrichment in $^{13}C$ (higher $\delta^{13}C$ values) with decreasing chain length between ca. 12,600 – 11,600 yrs BP. The highest $\delta^{13}C$ values (-28‰) were measured for $nC_{23}$ at ca. 11,900 yrs BP. Data tables with $n$-alkane concentrations and $\delta^{13}C$ values can be found in the Supplementary Tables S1 and S2.

## 5. Discussion

### 5.1. Sources of $n$-alkanes

#### 5.1.1 Potential and constraints of $n$-alkane based proxies

Sedimentary leaf wax $n$-alkane sources can be broadly distinguished based on $n$-alkane chain length. Principally, mid-chain alkanes ($nC_{23}$ and $nC_{25}$) are mostly attributed to aquatic sources or *Sphagnum* species while long-chain compounds can be derived from terrestrial plants or emergent macrophytes (Eglinton and Calvin, 1967; Ficken et al., 2000; Baas et al., 2000). In addition, it has been postulated that $nC_{31}$ is indicative of grasses as trees synthesize preferably $nC_{27}$ and $nC_{29}$ alkanes. Hence ratios of these compounds, such as $nC_{27}$ vs. $nC_{31}$ have frequently been applied in sedimentary archives to infer





ecological changes in lake catchments such as shifts from grassland to trees (Meyers and Ishiwatari, 1993; Schwark et al. 2002; Meyers, 2003).

Nevertheless an increasing number of studies indicated that alkane-patterns of plants are heterogeneous and compound distributions often overlap. For example, in some environments so-called "aquatic" mid-chain compounds ($nC_{23}$ and $nC_{25}$)

have been shown to be derived partly from terrestrial sources (Aichner et al., 2010; Gao et al., 2011), while long-chain "terrestrial" compounds ($nC_{27}$ and $nC_{29}$) were derived mainly from aquatic sources (Liu et al., 2015). Further, compilations of global plant data sets have shown that indices such as ACL and long-chain $n$-alkane ratios do not reliably reflect classes of terrestrial vegetation (Bush and McInerney, 2013). In addition, the sedimentary $n$-alkane pattern will be dominated by plants producing and delivering the largest amount of $n$-alkanes into the sedimentary sink. Based on a large-scale survey of

different tree species, Diefendorf et al. (2011) found that angiosperms showed a tendency to produce higher concentration of $n$-alkane leaf wax lipids, a result later confirmed by Bush and McInerney (2013).

### 5.1.2. Carbon isotopes as source indicators

Further information for source assessment of the different $n$-alkane homologues can be derived from their carbon isotopic

signature. For instance, in west African lakes with catchments dominated by $C_4$-vegetation, only $nC_{31}$ alkanes did show a $C_4$ isotopic signal. In contrast other long-chain $n$-alkanes carried a $C_3$-plant signature (Garcin et al. 2014), likely resulting from higher $n$-alkane production in the few $C_3$-trees in the lake catchment, compared to the dominating $C_4$-grasses. While no $C_4$-plants were present in eastern Europe, aquatic plants are also characterized by more positive $\delta^{13}C$ values (Allen and Spence, 1981; Keeley and Sandquist, 1992; Ficken et al., 2000; Mead et al., 2005; Aichner et al., 2010a), and an aquatic origin of

mid- and long-chain $n$-alkanes can be confirmed or declined: the constant $\delta^{13}C$ values of ca. -33‰ for $nC_{31}$ and slightly more variable values for $nC_{29}$ in our data set from TRZ paleolake suggest a stable terrestrial $C_3$-vegetation source for these compounds (Meyers, and Ishiwatari, 1993, Meyer, 2003) during the study period. The $^{13}C$ enrichment in $nC_{23}$, $nC_{25}$ and $nC_{27}$ during the YD could be explained by a) a change of sources of these compounds; or b) limited $CO_2$ availability to potential aquatic sources. The latter could be induced by low atmospheric $pCO_2$ or by establishment of diffusion barriers such as ice

cover and/or a thickened boundary layer in stagnant water. $CO_2$-limitation due to enhanced aquatic productivity, higher water temperature, salinity, or pH can be excluded (Street-Perrott et al., 2004; Aichner et al., 2010a,b); More positive $\delta^{13}C$ values could also result from a reduced isotopic fractionation of terrestrial plants subjected to water stress (Farquhar 1989), in particular during the YD, where 8-15% lower humidity have been reconstructed at MFM (Rach et al., 2014; 2017). On the other hand, increased aridity would affect all terrestrial plants and thus should be reflected in all $n$-alkane homologues,

contrary to our observations.



### 5.1.3 Comparison with palynological proxies

*Mid-chain n-alkanes (nC$_{23}$, nC$_{25}$)*

Concentrations of mid-chain length compounds were low throughout the whole record. For instance, $n$C$_{23}$, exceeds 10 µg / g

d.w. only during some episodes in the Allerød. (Fig. 2). Submerged aquatic macrophytes principally produce low amounts of pollen and mainly use vegetative strategies for reproduction. Consequently, very low concentrations of pollen have been counted in the sediment samples. However, also macrofossil-remains of submerged species *Potamogeton* and *Chara* have been found only in low amounts in some samples from the Allerød and early YD (Fig. 3). This gives evidence that submerged aquatic species have been of low abundance and contributed little to sedimentary organic matter in TRZ

throughout the studied time interval, with a somewhat higher proportional input during the Allerød and early YD. Low $\delta^{13}$C values, which vary between -33‰ and -28‰, also indicate a primary terrestrial origin of mid-chain $n$-alkanes. Submerged macrophyte use $^{13}$C-enriched bicarbonate as carbon source, especially when carbon is limited at highly productive stands, leading to $\delta^{13}$C-values of $n$-alkanes which can reach values up to -12‰ (Allen and Spence, 1981; Keeley and Sandquist, 1992; Aichner et al., 2010 a, b). Hence, low $\delta^{13}$C values for mid-chain compounds as measured in TRZ either indicate a) low

productivity of submerged macrophytes and/or b) relatively high proportional contribution of terrestrial sources to $n$C$_{23}$ and $n$C$_{25}$. We conclude that these mid-chain compounds comprise a mixture of aquatic and terrestrial sources during the Allerød and YD-onset, while during YD and Holocene $n$C$_{23}$ and $n$C$_{25}$ can be interpreted as of mainly terrestrial origin.

*Long-chain n-alkanes (nC$_{27}$, nC$_{29}$, nC$_{31}$)*

To further elucidate plant sources of $n$-alkanes in the sediments of TRZ we compare our organic geochemical data to pollen and macrofossil spectra (Fig. 3; low resolution pollen data from Wulf et al., 2013; decadal pollen/macrofossil data across the Allerød/YD-transition from Słowinski et al., 2017; unpublished low resolution macrofossil data from 12,400 - 10,000 yrs BP). Based on these data, the YD is characterized by relatively low amounts of tree and shrub pollen, except for *Juniperus* which expands after the YD-onset, and enhanced proportional input from herbaceaous plants (Poaceae, Cyperaceae,

*Artemisia,* Chenopodiaceae). The arboreal communities during the Allerød and Holocene were mainly dominated by *Pinus*. Relatively high amounts and percentages of pollen and macrofossils derived from *Betula* sp. were mainly observed in samples from the YD to Holocene transition and also in samples from the early Allerød (Fig. 3 and 4).

From the aliphatic compounds in TRZ, the $n$C$_{27}$-alkane shows the largest variability in concentration, which range from <10 µg / g d.w. during the YD up to >100 µg / g d.w. during the Allerød. This seems to reflect the trend of contribution of

arboreal pollen, specifically from *Betula*, to the lake sediments (Figs 3 and 7). Leaf waxes of the gymnosperm tree species *Pinus sylvestris* have been shown to contain often high relative abundances of $n$C$_{27}$ alkanes, but absolute concentrations are in most cases very low (Maffei et al., 2004; Ali et al., 2005; Dove and Mayes, 2005; Diefendorf et al., 2011; Bush and





McInerney, 2013). It is therefore likely that *Pinus* does not contribute significant amounts of *n*-alkanes to the sedimentary archive of TRZ. On the other hand, *Betula* sp. as the second most abundant tree species in the lake catchment could have been a major contributor to the sedimentary $n$C$_{27}$-pool. This species has also been reported to contain high relative abundances of those compounds (Schwark et al., 2002; Diefendorf et al., 2011), and likely biosynthesizes higher absolute

amounts, compared to *Pinus*. Comparing palynological abundances and $n$C$_{27}$ concentration data supports this idea: the high concentrations of $n$C$_{27}$-alkanes during the early Allerød, between ca. 13,250 and 13,100 yrs BP, coincide with an interval characterized by a dominance of *Betula* over *Pinus* (Wulf et al., 2013). Further, the increasing concentrations of $n$C$_{27}$ between ca. 11,700 and 11,500 yrs BP are in phase with *Betula* expansion at the YD/Holocene transition (Fig. 3). Finally, the expansion of *Pinus* after the onset of the Holocene is not reflected in increasing $n$C$_{27}$ concentrations, rather a decrease of *n*-

alkane concentrations was observed. This decrease can be explained by a synchronous decrease in absolute and relative *Betula* abundances. Based on the covariation of $n$C$_{27}$ concentrations and palynological data we consider *Betula sp.,* as a dominant contributor to these compounds while *Pinus sylvestris* appears to be a less relevant source.

Another gymnosperm shrub species, *Juniperus communis,* is mainly expanding during the early YD, peaking between ca. 12,300 and 12,500 yrs BP, before gradually declining (Fig. 3; Wulf et al., 2013; Słowinski et al., 2017). In the literature,

reported *n*-alkane concentrations for this species range from very low (Maffei et al., 2004) to high (Mayes et al, 1994). In the study of Diefendorf et al. (2011), *Juniperus osteosperma* is the only analysed gymnosperm which shows intermediate concentrations of *n*-alkanes. All these studies report, that *Juniperus* sp. biosynthesizes relatively long-chain lengths, in the range C$_{31}$-C$_{35}$, a pattern which has also been observed in plant samples from Lake Steisslingen (Southern Germany; Schwark et al., 2002). This suggests that *Juniperus* sp. could be a significant contributor to $n$C$_{31}$ also in the TRZ sediments, especially

during the early YD. During phases of high $n$C$_{31}$ abundance in the Allerød and Holocene other contributors must have been pre-dominant.

Herbaceous plants could theoretically contribute to all measured *n*-alkanes, but have often been associated with a dominance of longer *n*-alkane chain-lengths (e.g. Maffei, 1994, 1996; Zhang et al., 2004; Rommerskirchen et al., 2006). In TRZ, grasses and other herb pollen show varying concentrations, with a tendency of higher amounts during the YD. Their proportional

contribution significantly increases and decreases at the YD onset and termination and is enhanced by the strong decline of tree pollen during the cold interval. We assume that these types of plants indeed could be strong contributors to the longer chain lengths, i.e. $n$C$_{29}$ and $n$C$_{31}$, specifically during the YD when concentrations of those compounds approaching values of the dominating $n$C$_{27}$ alkane. A proportional contribution from trees, however, cannot be fully excluded, especially during the above mentioned phase of *Juniperus* expansion (early YD) as well as during parts of the early Holocene and Allerød when

*Betula* was dominating.

Considering emergent aquatic and telmatic species, macrofossils have been counted for species such as *Typha latifolium* and *Sparganium* as well as multiple species of ferns (Fig. 4). Those species might have contributed to the pool of long-chain *n*-alkanes (Ficken et al., 2000) at least during the Allerød and possibly also during the Holocene. After the YD-onset, those





species widely disappeared from the littoral due to their low tolerance of cool summer temperature (Fig. 4; Słowinski, et al., 2017).

*Statistical correlation across the Allerød/YD transition*

To analyze possible correlations between occurrences of alkanes with vegetation more in detail, we compared *n*-alkane concentrations throughout the Allerød and transition into the YD (ca. 13,350 – 12,400 yrs BP) with high-resolution pollen-data from the same interval (Fig. 4 and 5). For the latter, data published in Słowinski et al., 2017 (1230 – 1262 cm) have been complemented by further unpublished data extending further into the Allerød (1262 – 1271 cm). Macrofossils have been excluded from this survey a) due to lower sampling resolution (1 cm vs 0.5 cm for organic geochemical proxies)

compared to pollen and organic geochemical data and b) as macrofossil samples were not taken from the same core.

We found, that AP counts as well as some herbaceous plants and ferns showed a significant positive correlation with concentrations of $C_{25}$, $C_{27}$ and $C_{29}$ *n*-alkanes. In contrast, $nC_{31}$ concentrations showed a significant correlation with NAP counts, but also with *Salix*. Significant correlations are also found between *n*-alkane concentrations and algae (*Botryococcus, Pediastrum*) since the latter have been found in relatively high concentrations during the Allerød. Since *Botryococcus* does

not produce *n*-alkanes (Lichtfouse et al., 1994), this could well be considered as an autocorrelation.

It has to be noted that *n*-alkane concentrations strongly correlate with each other (Supplementary Table S3). This autocorrelation is due to the fact that basically all *n*-alkanes show the same rough trend with high concentrations during the Allerød and low concentrations during the YD. Autocorrelations can also be found within pollen data, e.g. *Pinus* counts significantly correlate with *Betula* (R = 0.79; p < 0.01; Supplementary Table S4). Hence, significant correlation does not

necessarily indicate origin of *n*-alkanes but could be due to intercorrelations between concentrations of source organisms, which is specifically relevant for negative correlations. Principally, the correlation coefficients reflect the major trends of pollen counts and *n*-alkane concentrations throughout the Allerød/YD-transition. Significant positive correlations indicate that the pollen counts follow the rough trend of *n*-alkanes, i.e. high concentrations during the Allerød, low concentration during the YD. In contrast, significant negative correlations indicate opposite trends, as relevant for *Juniperus* and some

herbs (*Artemisia,* Chenopodiaceae, *Helianthenum*)

We conclude that compounds can be attributed to arboreal and non-arboreal sources. Comparison with pollen data suggest, that the abundances of *Betula* and *Juniperus* in the catchment have a strong impact on *n*-alkane concentrations and composition of *n*-alkanes in TRZ sediments.   Decreasing concentrations of Mid-chain compounds, which are often interpreted as of aquatic origin, are here rather a mixture of aquatic and terrestrial sources, with high proportional input of

the latter during certain time periods (see above).



### 5.2. Potential of *n*-alkane ratios as paleoecological proxies in TRZ

Downcore changes of proxies based on proportional contribution of specific *n*-alkanes (e.g. $P_{aq}$, ACL, *n*-alkane ratios such as $nC_{27}/nC_{31}$, etc.) have frequently been applied to decipher paleoecological changes. Nevertheless, due to the above mentioned constraints, the applicability of alkane ratios as source indicators has been questioned, acknowledging that a uniform

interpretation of proxies such as ACL is not possible (Bush and McInerney, 2013; Rao et al., 2011, Hoffmann et al., 2013). However, our data illustrate that vegetation changes co-occur with compositional changes in sedimentary *n*-alkanes and we argue that if constraints from other proxy data can be made, specific additional information can be derived from such proxies.

To test their applicability in our study area we correlate standard *n*-alkane ratios with pollen percentages during the interval

13,350 – 12,400 yrs BP (Allerød/YD-transition) (Fig. 6), where the strongest changes occurred. Positive correlations were found between *n*-alkane ratios and arboreal pollen from *Pinus* and *Betula*. The same ratios showed negative correlations to non-arboreal pollen (NAP) and arboreal pollen (AP) from *Juniperus* and *Salix*. The highest positive PCC was given by the ratio $nC_{27} / (nC_{27} + nC_{31})$ versus percentage of $\Sigma$AP excluding *Juniperus* and *Salix* (R = 0.81, p < 0.001; Fig. 2 and Table 3). The same ratio also delivered most negative coefficients for correlations versus relative amounts of NAP, grasses, herbs,

*Juniperus* and *Salix*. Slightly lower correlations were found for the ratio $(nC_{27} + nC_{29}) / (nC_{27}+nC_{29}+nC_{31})$, but principally all the analysed ratios were good indicators to express changes of vegetation. Interestingly, the ratio $nC_{29} / (nC_{29} +nC_{31})$ strongly correlates with percentages of *Betula* (R = 0.80, p < 0.001; Table 3) which is considered as a major $nC_{27}$-producer. Given the discussion of sources of compounds above, this could indicate that this proxy is useful to assess proportional contribution of *Betula* to long-chain *n*-alkanes $C_{29}$ and $C_{31}$.

As discussed for pollen and *n*-alkane concentrations, autocorrelations need to be considered when interpreting these data. For instance, if aquatic contribution to the sediment is low as in TRZ, than the $P_{aq}$ must not be interpreted as a measure for aquatic influx but instead as an alternative expression for ACL, which explains the similarity of both curves (Fig. 2).

Although an exact assignment to sources is difficult, significant changes of *n*-alkane ratios mirror shifts of pollen percentages throughout the YD-onset. Thus we consider alkane ratios to be a good indicator for major ecological changes in

the lake catchment. We suggest that a shift towards longer chain-lengths at the YD-onset (reflected by higher ACL and by long-chain alkane ratios), is indicative for a shift of a more tree-dominated catchment (*Pinus/Betula* communities, Wulf et al., 2013) to a higher abundance of herbaceous plants, complemented by shrubs *Juniperus* and *Salix*.

### 5.3. Paleoecological and climatic context

During the Allerød, i.e. between 13,360 and 12,680 yrs BP an average ACL of ca. 26.9 is reflecting the above mentioned vegetation communities, consisting of *Pinus* and *Betula*-forests (Fig. 7). Shifts towards longer chain lengths (>27.4) occur between ca. 13,220 – 13,050  yrs BP, and 12,970 – 12,850 years BP. Climatic variations during the Allerød have been



detected in numerous paleoclimatic records. The most recent chronology from NGRIP ice cores dates the Greenland-Interstadial cooling phase 1b (GI-1b) to 13,261 ± 149 to 13,049 ± 143 yrs BP (Rasmussen et al., 2014; Fig. 5). This episode has been discussed to be probably equivalent to the "Gerzensee oscillation", which has been observed in lake sediment records (von Grafenstein et al., 1999, 2000). In our record the ACL reversal between ca. 13,220 – 13,050 yrs BP could

probably be considered as a response of vegetation in the TRZ catchment to this cold oscillation. Another minor and short-term cooling reversal ca. 40 yrs before the onset of the YD, at around 12,720 yrs BP, has been recently proposed for MFM (Engels et al., 2016) but is not clearly recorded by *n*-alkane ratios in TRZ.

The sharpest and most pronounced decrease of ACL occurs between 12,680 and 12,600 yrs BP. This is synchronous with the transition phase of major vegetation changes as inferred from palynological data between 12,680 and 12,620 yrs BP

(Słowinski et al., 2017). Hence, no pronounced lag-time for organic biomarkers, for instance due to prolonged residence time in soils or transport from the catchment to the lake can be observed. Similar to more western locations, like MFM, the rapid vegetation change which marks the YD onset occurred with a delay of ca 170 years compared to the start of cooling as inferred from Greenland ice cores (GS-1-onset 12,846 ± 138 yrs BP; Rasmussen et al., 2014).

Within the YD cold interval, the ACL is showing a gradual trend towards lower values. While some potential contributors to

longer chain lengths such as *Poaceae*, *Cyperaceae* and other herbaceous plants show relatively constant pollen counts during the YD, some others such as *Juniperus* are more abundant during the first half of the YD (Wulf et al., 2013; Fig. 3 and 7). This is in agreement with the principal trend of $\delta^{18}O$ values and $Ca^{2+}$-concentrations in Greenland ice cores, which indicate coldest and driest conditions in the earlier phase of the YD, followed by a gradual and slight warming (Rasmussen et al., 2014). A similar pattern has also been detected within proxy data from other European lakes like, for example, Lake

MFM (varve thickness, elemental composition and δD values; Brauer et al., 1999a,b; Rach et al., 2014), Lake Ammersee ($\delta^{18}O$; von Grafenstein et al., 2003), palaeolake Rehwiese (varve thickness; Neugebauer et al., 2012), and Lake Kråkenes (elemental composition; Lane et al., 2013; Fig. 6). Recently this bi-partitioning was also observed in Lake Suigetsu (Schlolaut et al., 2017). Explanations for this phenomenon have been related to a re-strengthening of the AMOC at ca. 12,300 yrs BP, whose slowdown and consequently southward shifting of the Polar front are considered as a major trigger for the

YD in Europe (McManus et al., 2004; Bakke et al., 2009; Elmore and Wright, 2011). The gradual strengthening of the AMOC pushed the Polar front back to the North, which is the reason that the partially rapid climatic responses occurred earlier at more southern (ca. 12,240 yrs BP at MFM) compared to more northern Locations (ca. 12,140 yrs BP at Lake Kråkenes) (Brauer et al., 2008; Lane et al., 2013). At TRZ, ACL, as an integrative proxy, shows shifts at ca. 12,300-12,280 and 12,200-12,190 yrs BP, but changes are gradual and do not allow a distinct timing of this climatic change. An explanation

for this could be the continental setting of our study area, which show weaker climatic responses compared to locations with stronger Atlantic influence (Słowinski et al., 2017).

The succession of *n*-alkane abundances between ca, 11,540-11,200 yrs BP reflects the gradual expansion of vegetation around the YD/Holocene transition, defined by means of biostratigraphy at 11,515 ± 35 yrs BP (Table 1), and during the





early Holocene. An increase of $nC_{27}$ concentrations starting around 12,540 yrs BP, while $nC_{31}$ reaches the lowest concentrations within the studied interval, is possibly caused by expansion of *Betula* and triggers a sharp decrease of ACL between 11,580-11,490 years BP. This shift occurs ca. 70-80 years earlier compared to the defined onset of the Holocene in Greenland at 11,654 ± 4 yrs BP based on oxygen isotopes in the NGRIP ice core (Rasmussen et al., 2014). Due to age-

uncertainties between 35-90 yrs within the respective section of the TRZ-record, ultimate quantifications of the lag are not possible in this case.

A sudden increase of $nC_{29}$ and $nC_{31}$ concentrations at ca. 11,430 yrs BP (Fig. 2) i.e. 90 years after the start of increasing $nC27$ concentration is associated to a ca. 180 year reversal of ACL and *n*-alkane ratios, whose start is in phase with a cold period in Greenland, referred to as 11.4 kyr event (Rasmussen et al., 2014). In contrast to the YD, which is characterized by

low concentrations of *n*-alkanes, the increasing ACL at 11,5 kyrs BP is triggered by an increase of concentrations of long-chain compounds. Hence, it is possible that the rapid ecological developments in the lake catchment in course of warming, such as expansion of forest vegetation and development of soils, are the main drivers behind the observed shift and not a response to a cooling within the catchment.

Cold reversals during the early Holocene are mostly referred to as Preboreal Oscillations (PBO), have been inferred from

different proxies in many European lakes. The exact timing and possible synchronicity of these oscillations is still a matter of debate and PBOs are generally difficult to detect in lake sediments (e.g. Björk et al., 1997; Bos et al., 2007). Based on sedimentary analysis of a lake in southern Sweden, Wohlfahrt et al. (2006) suggested a PBO could be stratigraphically located between tephra layers Hässeldalen and Askja-S, which have been dated to 11,380 ± 216 and 11,228 ± 226, respectively (Ott et al., 2016). The fact that the decrease of ACL in the TRZ sediment core is observed ca. 10 cm below the

Askja-S layer is evidence against a connection to the oscillation at least according to the stratigraphic boundaries suggested by Wohlfahrt et al. (2006).

The upper 31 cm of the studied core interval cover the early Holocene from ca, 10,970 to 9,940 yrs BP at lower resolution due to lower sedimentation rates and increased sample size. This episode is characterized by a gradual change of *n*-alkane ratios, e.g. decrease of ACL, indicating the gradual establishment of a Holocene vegetation in the lake catchment due to

further gradual warming. Specifically after ca. 10,600 yrs BP, a decrease of mainly $nC_{27}$-alkane concentrations probably indicates a decline of *Betula* and further expansion of *Pinus* as dominating species (Wulf et al., 2013).

## 6. Conclusion

The YD cold interval is characterized by relatively low concentrations of *n*-alkanes and pronounced changes of relative

abundances of *n*-alkanes, which result in a strong increase of ACL, and changes of *n*-alkane ratios. The shift towards higher ACL at the Allerød/YD transition between 12,680 and 12,600 yrs BP is in phase with a lower contribution of tree pollen



(*Pinus* and *Betula*) to the sediments, and a subsequently higher abundance of pollen derived from herbaceaous plants (*Poaceae*, *Cyperaceae* and *Artemisia*), as well as shrubs (*Juniperus* and *Salix).* These major vegetation changes occurred ca. 170 yrs after the start of $\delta^{18}$O-decrease (GS-1-onset 12,846 ± 138 yrs BP) in Greenland ice-cores. The YD/Holocene-transition is characterized by a gradual increase of $n$C27-concentrations starting at ca. 11,540 – 11,530 yrs BP, most likely

induced by an expansion of *Betula*, and a sharp increase of $n$C29- and $n$C31-concentrations ca. 90 years later. Due to larger age-uncertainties in the non-laminated section of the core, these ecological changes are more difficult to be set into context with the major warming in Greenland (11,654 ± 4 yrs BP).

Short term reversals of ACL and $n$-alkane ratios during the Allerød could possibly be linked to GI-1b, while a similar pattern during the early Holocene is more likely to have been triggered by rapid ecological responses in course of warming. These

results show that there is no standard interpretation of changing proxy values. Instead, responses such as changes of ACL need to be discussed under careful evaluation of the parameters which lead to the observed changes. Taking our studied interval, the major identified shifts towards longer chain lengths (i.e. during the Allerød, the YD, between 11,430 – 11,250 yrs BP, and after 11,070 years) have had different causes, because either increase or decrease of specific compounds trigger the individual response.

We did not observe significant lag times between the response of pollen and $n$-alkane data during periods of abrupt change (i.e. on multidecadal timescales), indicating $n$-alkane residence times in the catchment below our sampling resolution (decades). However, organic geochemical data integrate different information compared to pollen, as there is evidence that *Pinus sylvetris*, despite being a dominant tree species in the TRZ catchment, has a minor impact on $n$-alkane based parameters. Overall, this study shows that ACL and ratios of $n$-alkanes are suitable integrative proxies to track major and

abrupt vegetation changes in a local setting.

**Acknowledgements:**

This study was supported through an ERC Consolidator Grant (STEEPclim, Grant agreement No. 647035) to D.S., by a grant from the Polish National Science Centre (2015/17/B/ST10/03430) to M. S., and a 6 month fellowship from the German Academic Exchange Service (DAAD) to B.A.. We thank Sophie Boven and Michael Poehle for help during sample

preparation.

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





**Tables**

**Table 1:** Age-anchor points used for age-depth calibration of the studied core section (Słowinski et al 2017). * mark omitted ages. LST: Laacher See Tephra. Age from Brauer et al. (1999a). Askja-S age from Ott et al. (2016). Biostratigraphic age at YD/Holocene transition from Ralska-Jasiewiczowa et al. (1992) and Litt et al. (2001). Age-depth plot of age-model in

5    Supplementary Figure 1.

| Lab. Code | Composite depth [cm] | Dated material / age-anchor point | AMS [14]C age (yrs BP) | Calibrated age (cal yrs BP±2σ) | Modelled age (cal yrs BP±2σ) |
|---|---|---|---|---|---|
| Poz-39366 | 1145 | *Pinus* needle and bud scales, *Betula* sp. fruits and bud scales | 8970 ± 50 | 10076 ± 158 | 10094 ± 150 |
| GdA-3008 | 1156 | *Betula* sp. fruits and bud scales, *Pinus* bud scales | 9405 ± 38 | 10629 ± 106 | 10625 ± 101 |
| - | 1164 | varve counted age of Askja-S Tephra in Lake Czechowskie | | 11228± 226 | |
| GdA-3009 | 1165 | Pinus needle and bud scales, *Betula* sp. fruits and bud scales | 9587 ± 38 | 10938 ± 187 | 10934 ± 169 |
| Poz-39367 | 1176 | *Betula* sp. fruits and bud scales, *Pinus* needle and bud scales | 9970 ± 60 | 11474 ± 231 | 11377 ± 156 |
| - | 1183.5 | biostratigraphic change; age information from Lake Gosciaz | | 11515 ± 35 | |
| GdA-3010* | 1241 | *Pinus* bud scales, *Betula* sp. fruits and bud scales | 10824 ± 42 | 12726 ± 47* | |
| GdA-3011* | 1244 | *Betula* sp. fruits and bud scales, *Pinus* needle and bud scales | 10950 ± 45 | 12831 ± 120* | |
| - | 1244.5 | end varved sediments | | 12678 ± 43 | |
| - | 1253.75 | varve counted age of LST in the MFM record | | 12880 ± 40 | |
| - | 1262 | begin varved sediments | | 13043 ± 43 | |
| GdA-3012* | 1263 | *Pinus* needle and bud scales, *Betula* sp. fruits and bud scales | 11051 ± 44 | 12920 ± 131* | |
| Poz-39369* | 1269 | *Pinus* needle and bud scales, *Drysa octopetala* - leaves, *Betula* sp. fruits and bud scales | 12040 ± 70 | 13914 ± 169* | |
| GdA-3013 | 1276 | *Pinus* needle, *Betula* sp. fruits and bud scales, *Betula* nana leaves | 11718 ± 47 | 13576 ± 137 | 13522 ± 94 |



**Figures**

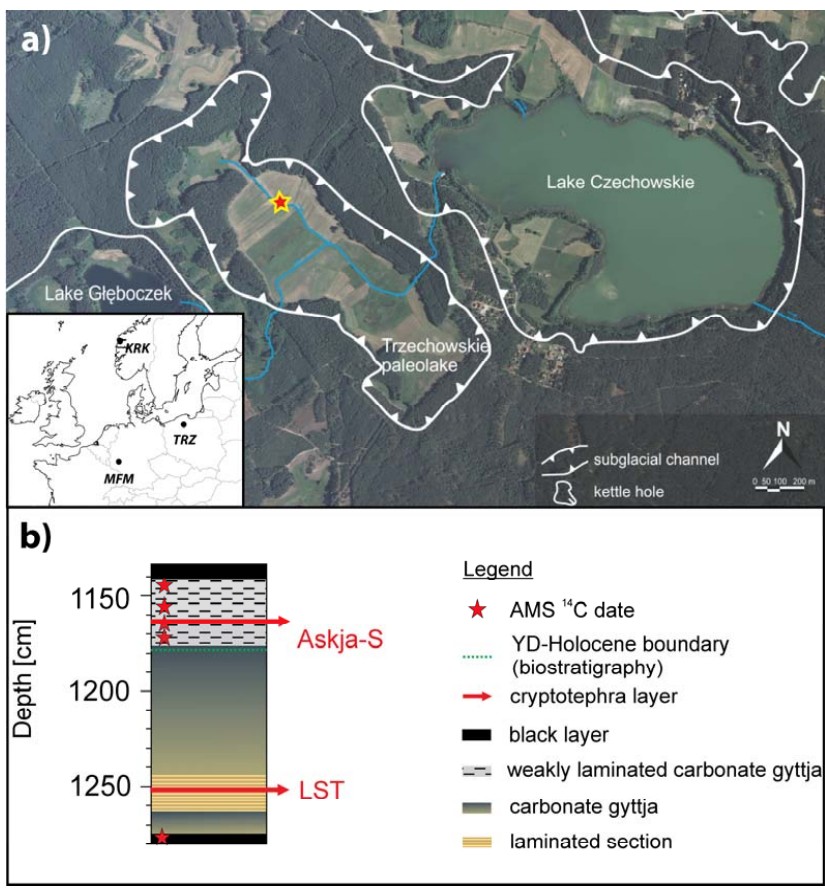

5   **Figure 1:** a) Coring position (red-yellow asterisk) at Trzechowskie paleolake (TRZ). b) Schematic overview over sampled
     core section with anchor points used for the age-model. LST: Laacher See Tephra. MFM: Meerfelder Maar. KRK: Lake
     Kråkenes.





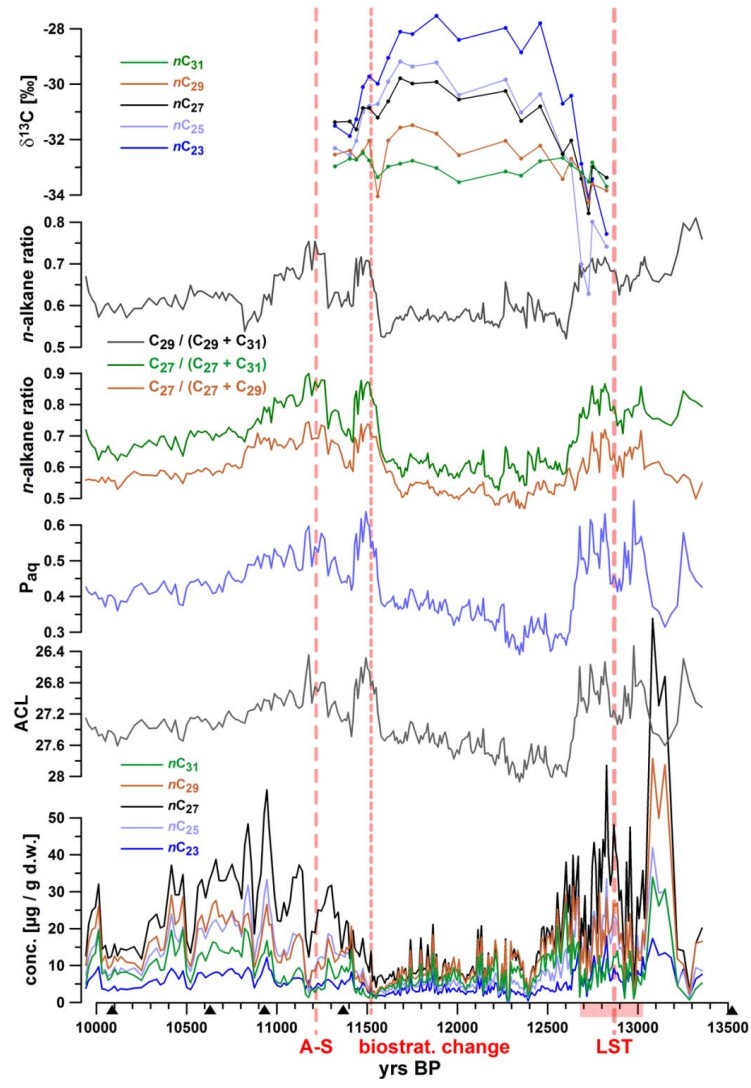

**Figure 2:** Concentrations of *n*-alkanes, average chain lengh (ACL), P$_{aq}$, *n*-alkane ratios, as well as δ$^{13}$C values plotted versus age. Red dotted lines mark occurrences of tephra-layers Askja-S (A-S: 11,228 ± 226 yrs BP) and the Laacher See Tephra (LST 12,880 ± 40 yrs BP), and the biostratigraphic change at 1183.5cm depth (11,515 ± 35 yrs BP). Black triangles mark

5    $^{14}$C-AMS-ages. The pink shaded interval on the x-axis illustrates the laminated core-section.





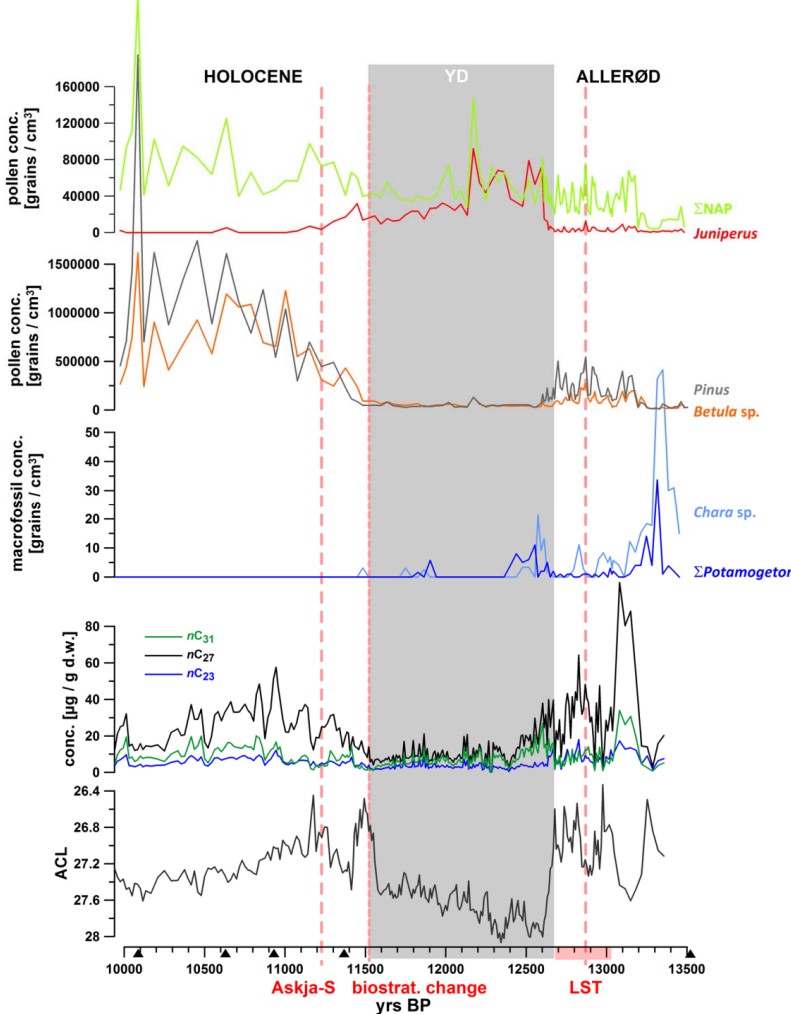

**Figure 3:** *n*-Alkane concentrations and average chain lengths (ACL) during the studied interval in comparison to pollen and macrofossil counts (Wulf et al., 2013; Słowinski et al., 2017; unpublished macrofossil data from 10,000 – 12,400 yrs BP). NAP: non-arboreal pollen. Grey-shaded interval: YD according to pollen-cluster.



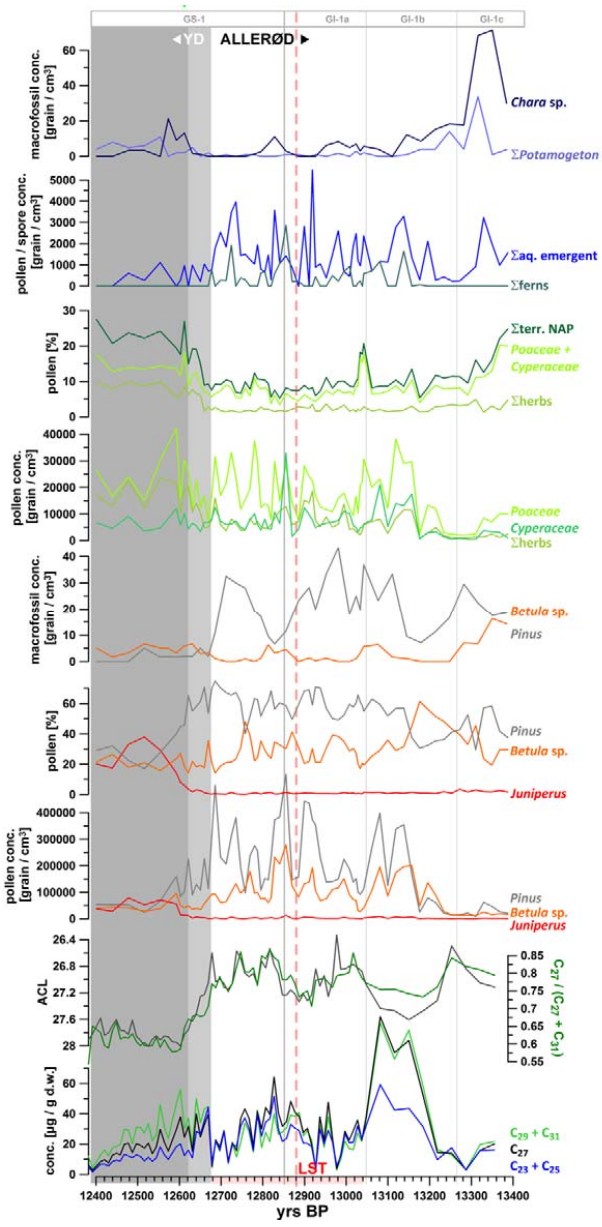



**Figure 4:** High resolution palyonological data from the Allerød and transition into the YD in comparison to concentration of *n*-alkanes, ACL and the $C_{27}$ / ($C_{27}$+$C_{31}$) *n*-alkane-ratio. The light grey shaded period indicates the transition from the Allerød to the YD (12,680 – 12,620 yrs BP) as defined by Słowinski et al. (2017), while the dark grey shaded interval marks the YD. NGRIP ice core stratification according to Rasmussen et al. (2014). Σherbs: NAP without Poaceae and Cyperaceae.

|  | c C23 | c C25 | c C27 | c C29 | c C31 | Σ all | Σ 23 25 | Σ 29 31 | Σ 27 29 31 |
|---|---|---|---|---|---|---|---|---|---|
| c Pinus sylvestris | 0.37 | 0.49 | 0.43 | 0.34 | 0.27 | 0.41 | 0.46 | 0.32 | 0.38 |
| c Salix | 0.24 | 0.27 | 0.46 | 0.53 | 0.60 | 0.47 | 0.26 | 0.56 | 0.52 |
| c Betula | 0.37 | 0.38 | 0.39 | 0.30 | 0.19 | 0.35 | 0.39 | 0.27 | 0.33 |
| c Hippuris rhamn. | 0.34 | 0.24 | 0.35 | 0.39 | 0.32 | 0.35 | 0.27 | 0.37 | 0.37 |
| c Juniperus | -0.47 | -0.40 | -0.27 | -0.16 | 0.05 | -0.24 | -0.42 | -0.09 | -0.18 |
| c Σ AP | 0.34 | 0.45 | 0.42 | 0.34 | 0.28 | 0.40 | 0.42 | 0.32 | 0.38 |
|  |  |  |  |  |  |  |  |  |  |
| c Calluna vulgaris | 0.14 | 0.15 | 0.06 | -0.01 | -0.04 | 0.05 | 0.15 | -0.02 | 0.02 |
| c Ericaceae | -0.26 | -0.21 | -0.21 | -0.22 | -0.07 | -0.20 | -0.23 | -0.17 | -0.19 |
| c Helianthenum | -0.45 | -0.41 | -0.32 | -0.28 | -0.19 | -0.33 | -0.43 | -0.25 | -0.29 |
| c Dryas octopetala | 0.21 | 0.10 | 0.18 | 0.23 | 0.18 | 0.19 | 0.13 | 0.21 | 0.20 |
| c Ranunculus | 0.05 | 0.01 | 0.00 | -0.04 | -0.06 | -0.01 | 0.02 | -0.05 | -0.03 |
| c Rumex | -0.17 | -0.18 | 0.00 | 0.07 | 0.19 | 0.00 | -0.18 | 0.12 | 0.06 |
| c Filipendula | 0.16 | 0.32 | 0.24 | 0.15 | 0.11 | 0.22 | 0.27 | 0.14 | 0.19 |
| c Thalictrum | 0.32 | 0.40 | 0.46 | 0.41 | 0.40 | 0.44 | 0.38 | 0.41 | 0.44 |
| c Artemisia | -0.51 | -0.38 | -0.27 | -0.17 | 0.11 | -0.24 | -0.43 | -0.07 | -0.17 |
| c Chenopodiacea | -0.44 | -0.30 | -0.27 | -0.22 | 0.01 | -0.24 | -0.35 | -0.14 | -0.21 |
| c Poaceae | -0.04 | 0.06 | 0.15 | 0.20 | 0.32 | 0.17 | 0.03 | 0.25 | 0.20 |
| c Cyperaceae | 0.23 | 0.32 | 0.39 | 0.37 | 0.36 | 0.37 | 0.30 | 0.37 | 0.39 |
| c Σ terr. NAP | -0.12 | 0.01 | 0.14 | 0.19 | 0.35 | 0.15 | -0.02 | 0.25 | 0.19 |
|  |  |  |  |  |  |  |  |  |  |
| c Typha latifolia | -0.03 | -0.01 | -0.07 | -0.09 | -0.13 | -0.07 | -0.02 | -0.10 | -0.09 |
| c Equisetum | 0.26 | 0.23 | 0.17 | 0.13 | 0.01 | 0.16 | 0.24 | 0.09 | 0.13 |
| c Filicales monolete | 0.34 | 0.41 | 0.39 | 0.31 | 0.22 | 0.36 | 0.39 | 0.28 | 0.34 |
| c Botryococcus | 0.33 | 0.49 | 0.47 | 0.42 | 0.41 | 0.46 | 0.45 | 0.42 | 0.45 |
| c Pediastrum | 0.37 | 0.18 | 0.28 | 0.28 | 0.14 | 0.26 | 0.24 | 0.23 | 0.26 |

**Figure 5:** Heatmap table, illustrating Pearson correlation coefficients between concentrations of pollen and *n*-alkanes at the Allerød/YD-transition (1230 – 1271 cm). Data have been z-transformed before correlation. Bold black numbers: p < 0.01. Black numbers: p <. 0.05. Positive correlations in green and negative correlations in red. AP: arboreal pollen. NAP: terrestrial non-arboreal pollen.




| | P$_{aq}$ | C27 / (C27+C31) | C27 / (C27+C29) | C27 / (C27+C29+C31) | C29 / (C29+C31) | (C27 +C29) / (C27+C29+C31) | ACL |
|---|---|---|---|---|---|---|---|
| Σ AP | 0.69 | 0.78 | 0.65 | 0.76 | 0.59 | 0.74 | -0.70 |
| Σ AP wo. Junip. | 0.68 | 0.80 | 0.62 | 0.75 | 0.63 | 0.77 | -0.72 |
| Σ AP wo. Junip. & Salix | 0.69 | 0.81 | 0.62 | 0.76 | 0.63 | 0.78 | -0.72 |
| Betula | 0.37 | 0.56 | 0.08 | 0.32 | 0.69 | 0.60 | -0.50 |
| Pinus sylvestris | 0.51 | 0.50 | 0.64 | 0.62 | 0.22 | 0.43 | -0.45 |
| Juniperus | -0.64 | -0.76 | -0.56 | -0.70 | -0.61 | -0.74 | 0.68 |
| Salix | -0.71 | -0.65 | -0.64 | -0.70 | -0.43 | -0.60 | 0.68 |
| Σ terr. NAP | -0.70 | -0.81 | -0.65 | -0.77 | -0.62 | -0.77 | 0.73 |
| Poaceae + Cyperaceae | -0.60 | -0.68 | -0.56 | -0.67 | -0.51 | -0.65 | 0.61 |
| Poaceae | -0.57 | -0.65 | -0.57 | -0.66 | -0.47 | -0.62 | 0.59 |
| Cyperaceae | -0.44 | -0.48 | -0.29 | -0.41 | -0.45 | -0.48 | 0.45 |
| Σ herbs | -0.72 | -0.85 | -0.65 | -0.80 | -0.68 | -0.82 | 0.76 |
| Artemisia | -0.70 | -0.85 | -0.62 | -0.78 | -0.70 | -0.83 | 0.75 |
| Chenopodia | -0.53 | -0.66 | -0.45 | -0.59 | -0.56 | -0.65 | 0.57 |
| Σ ferns | 0.23 | 0.38 | -0.04 | 0.15 | 0.52 | 0.43 | -0.31 |
| Σ aq subm & float | -0.21 | -0.23 | -0.27 | -0.27 | -0.11 | -0.20 | 0.21 |
| Σ aq emergent | 0.05 | 0.12 | -0.07 | 0.01 | 0.19 | 0.15 | -0.08 |
| Σ algae | 0.01 | 0.20 | -0.35 | -0.12 | 0.48 | 0.28 | -0.13 |
| Sphagnum | 0.46 | 0.38 | 0.57 | 0.53 | 0.13 | 0.32 | -0.42 |

5 **Figure 6:** Heatmap table, showing Pearson Correlation Coefficients between pollen percentages and *n*-alkane-ratios. Data have been z-transformed before correlation. Bold black numbers: p < 0.01. Black numbers: p <. 0.05. Positive correlations in green and negative correlations in red. Intercorrelation between relative abundances of pollen and *n*-alkane ratios are in the Supplementary Tables S3 and S5. AP: arboreal pollen. Terr. NAP: terrestrial non-arboreal pollen. Σ other herbs: NAP without Poaceae and Cyperaceae.



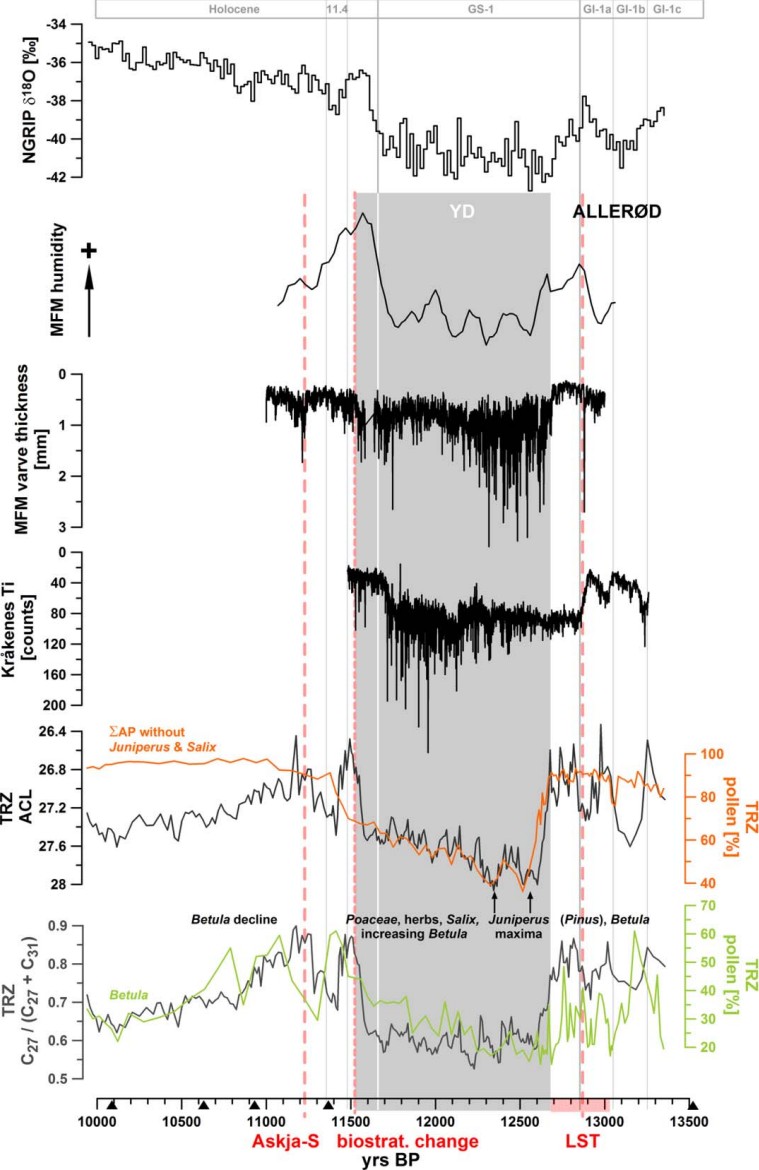

**Figure 7:** Data comparison to MFM (Brauer et al., 1999b; Rach et al., 2014), Lake Kråkenes (Bakke et al., 2009), and NGRIP (Rasmussen et al., 2014).