# Peer review of "Leaf wax *n*-alkane distributions record ecological changes during the Younger Dryas at Trzechowskie paleolake (Northern Poland) without temporal delay"

_Climate of the Past, 2018_

## Referee Comment (RC1) · Anonymous Referee #1 · 23 Apr 2018

Review of Manuscript CP-2018-6 entitled "Leaf wax n-alkane distributions record eco-
logical changes during the Younger Dryas at Trzechowskie paleolake (Northern Poland)
without temporal delay"

General Comments:

Overall, I think the manuscript is an important contribution to the fields of organic geo-
chemistry and paleoenvironmental reconstruction. The conclusions regarding the risks
associated with using oversimplified alkane-based metrics to reconstruct vegetation

and the lack of a delay between the alkane and pollen proxies are important to the broader discipline. The issue of 'pre-aging' of waxes is particularly troublesome in lacustrine settings and I think it deserves more attention in the discussion as it is currently limited to just a couple of paragraphs. I recommend adding references to this section including:

Gierga, M., Hajdas, I., van Raden, U.J., Gilli, A.,Wacker, L., Sturm, M., Bernasconi, S.M., Smittenberg, R.H., 2016. Long-stored soil carbon released by prehistoric land use: evidence from compound-specific radiocarbon analysis on Soppensee lake sediments. Quat. Sci. Rev. 144, 123e131.

Lane, C.S., Horn, S.P., Taylor, Z.P., Kerr, M., 2016. Correlation of bulk sedimentary and compound-specific d13C values indicates minimal pre-aging of n-alkanes in a small tropical watershed. Quat. Sci. Rev. 145, 238e242.

Uchikawa, J., Popp, B.N., Schoonmaker, J.E., Zu, L., 2008. Direct application of compound-specific radiocarbon analysis of leaf waxes to establish lacustrine sediment chronology. J. Paleolimn 39, 43e60.

The document would generally benefit from improved conciseness and clarity, this includes the abstract that is much too long for such a short paper.

It is my opinion that the study is worthy of publication in Climate of the Past, but considerable effort will be required to improve the conciseness of the presentation and to focus the paper more effectively on the most significant conclusions (lack of diagnostic capability based on widely-applied chain-length metrics and rapid response of alkane proxies to vegetation change). Too much of the

General editing that should be applied to the entirety of the text:

1. The use of conjunctive adverbs (however, thus, nevertheless, etc.) are excessive throughout the text. The paper would be much more concise if sentences were restructured to omit the conjunctive adverbs altogether. 2. The authors are flipping between

active and passive voice throughout the document. (Lots of examples, but see lines 28-30 on page 7 for one example: switch from 'have been reconstructed' to 'our observations'. 3. Throughout, commas should be placed before 'which' and 'but'. 4. Compound-specific needs to be hyphenated when used as an adjective. 5. The word 'this' is used too frequently in sentence strings where it is often difficult to decipher what precisely 'this' is referring to (e.g. page 2, line 28; 6. Shouldn't results and methods be in the past tense? 7. Throughout, need to be careful when designating species vs. genera. For example, page 9 Betula spp. would be a genus, not a species.

Specific comments:

1. Page 5, line 9: grammar 'hexane as solvent' 2. Page 5, line 20: Delta V Plus should be spaced 3. Page 5, line 26: should this be a permille sign? 4. Page 7, line 10: should be 'concentrations' 5. Page 7, line 14: great example of a sentence that could be much more concise. . .Source allocation of n-alkane homologues can also be derived from the carbon isotope composition. 6. Page 7, line 26: semicolon should be period. 7. Page 8, line 6-7: low concentrations of aquatic macrophyte pollen specifically? Needs to be specified. 8. Page 8, line 7: remove 'also' 9. Page 8, line 17: just say 'are likely of terrestrial origin'. 10. Page 8, line 25: remove 'mainly' 11. Page 9, line 5: no need for a colon here. 12. Page 9, line 13: recommend different word choice for 'mainly expanding' 13. Page 9, lines 14-16: no need for 'in the. . .' openings to both sentences 14. Page 10, line 16: need to specify what exactly is being correlated; also change from 'each other' to 'one another'. 15. Page 10, no need to capitalize 'Mid-chain' 16. Page 11, line 20: specify which data. . .avoid 'as discussed'. 17. Page 12, line 3: awkward phrasing 'has been discussed to be probably equivalent'

Figure 2: I am not sure that overlaying all of the homologues in the bottom panel is effective. . .there are too many lines making it difficult to decipher the trends of any one individual homologue in the plot.

Figures 4 and 5: latin names need to be italicized.

---

## Referee Comment (RC2) · Anonymous Referee #2 · 17 May 2018

The manuscript "Leaf wax n-alkane distributions record ecological changes during the Younger Dryas at Trzechowskie paleolake (Northern Poland) without temporal delay" by Aichner et al. uses high-resolution records of n-alkane distribution and pollen, and lower resolution d13C analyses, during the last deglacial period to address whether n-alkanes reflect plant community changes. This is a well-written paper describing an impressively detailed dataset, and the data support the conclusions. This paper is a step towards developing quantitative paleohydrological reconstructions using compound-specific hydrogen isotopes, and as such is an important contribution to the literature.

An assumption that is widely made regarding leaf waxes is that mid-chain n-alkanes reflect aquatic plant productivity and long-chain n-alkanes reflect terrestrial plant productivity, with longest chain lengths potentially being produced by grasses and herbs. These assumptions are mainly based on analyses of modern plants. Very little work directly compares n-alkane distributions with plant ecosystem composition as reflected by different proxies in sediment archives, so this study is a nice test of that assumption. In addition, if one can use n-alkane chain lengths to determine or control for plant community changes when interpreting leaf wax isotope measurements, then the analyses are simpler and less expensive than analyzing pollen or macrofossils in addition to n-alkanes.

The authors use appropriate methods, which could be clarified with a couple minor text additions, detailed below. These methods and results support their two main conclusions: 1. Changing measures of chain length distributions may be influenced by multiple different factors, and should therefore be examined carefully, in terms of which individual n-alkane chain lengths are actually changing, and 2. "ACL and ratios of n-alkanes are suitable integrative proxies to track major and abrupt vegetation changes in a local setting."

Substantive comments:

1. Changes in pollen and n-alkanes in the same sediment samples are indisputable. The authors should be a bit clearer about age uncertainties when comparing their record to proxy records from other archives. For example: p 13 line 5: the authors mention a lag here, but do not mention a lag for previous intervals. The age model uncertainty varies throughout the record, but at all times in this record, the uncertainty is enough to influence timing & interpretations/comparisons with other records. p 12 line 5: The discussion of the timing of the longer ACL at 13.2 to 13.0 ka: age uncertainty at this time period is ±250 years or so (based on visual inspection of the age model in Supp Fig. 1). The age model uncertainty should be acknowledged in the discussion, as the age model uncertainties in each of the records means that the events can't be

assumed to be synchronous. p 12 line 9: Similarly, the conclusion about the timing of the YD onset 170 years after GS-1 onset: the age uncertainty at 12.6 ka in this record appears to be about $\pm150$ years, so the lag could mainly be due to age model uncertainty. This should be acknowledged in the text. If the authors have evidence to suggest that the lag at this site is real, this would be a good place to present that.

2. Carbon isotopes: Isotopic measurements can be influenced by instrument drift through time. In addition, isotopic measurements can be strongly influenced by linearity (i.e., the size of the individual peaks being measured) (Kornfeld et al., 2012). The linearity effect is stronger for smaller peaks. Because the authors report d13C data on n-alkanes of widely varying concentrations in the same sample, these peaks are likely subject to strong changes in linearity, especially the smallest peaks (C23, which differs the most from the other n-alkane chain lengths in its d13C value). How do the authors account for drift and linearity effects in their d13C analyses? Do they run standards at varying concentrations and correct for the linearity effect? This is important to state, as these effects can dramatically impact isotope values. If the authors do not control for the effects of drift and linearity, they should also state that, as that limits the degree to which the values can be interpreted.

3. I have a question about each conclusion that would be helpful to clarify in the text: Conclusion 1: As far as I can understand from these results, it seems as if taking this type of data the next step and making a quantitative interpretation of compound-specific hydrogen isotope ratios still requires an independent record of plant ecology (i.e., pollen or macrofossils). Simply knowing, for example, that ACL increased because the long chain lengths increased doesn't allow for an interpretation of the ecological shift, as this shift could have been caused by one of several different plants in an ecosystem. Is that correct? If so, it could be helpful to clarify that independent ecosystem reconstructions will still be required for future quantitative isotope interpretations. If not correct, then the opposite can be clarified in the text. Conclusion 2: This dataset is a really nice illustration of the similar timing of pollen and n-alkane concentrations. These data do

not address whether ecological changes lag climate changes, correct? Perhaps the authors could point that out in the conclusion, as a reader may be inclined to interpret the data as such.

Minor comments:

Fig 1 a: what are the blue lines? I'm confused by the legend: it seems like the white lines with triangles are the kettle holes, but they're listed in the key as subglacial channels? I don't see the symbol for a kettle hole anywhere on the map. Can the symbols be clarified?

p 13 line 1: I think the authors mean 11,540 years ago

p 13 line 24: ACL is increasing through this Early Holocene interval, not decreasing, as stated in the text

Reference: Kornfeld, A., Horton, T. W., Yakir, D., & Turnbull, M. H. (2012). Correcting for nonlinearity effects of continuous flow isotope ratio mass spectrometry across a wide dynamic range. Rapid Communications in Mass Spectrometry, 26(4), 460–468. https://doi.org/10.1002/rcm.6120

---

## Referee Comment (RC3) · Anonymous Referee #1 · 4 Jun 2018

Too much of the manuscript focuses on paleoenvironmental/paleoclimate interpretations that have already been made based on pollen-based interpretations in prior studies when the truly novel aspects of this manuscript seem to be the rapid response of the alkane record to change relative to other studies that highlight 'pre-aging'.

---

## Author Comment (AC1) · 15 Jun 2018

General Comments: Overall, I think the manuscript is an important contribution to the fields of organic geochemistry and paleoenvironmental reconstruction. The conclusions regarding the risks associated with using oversimplified alkane-based metrics to reconstruct vegetation and the lack of a delay between the alkane and pollen proxies are important to the broader discipline. The issue of 'pre-aging' of waxes is particularly troublesome in lacustrine settings and I think it deserves more attention in the discussion as it is currently limited to just a couple of paragraphs. I recommend adding

references to this section including: - Gierga, M., Hajdas, I., van Raden, U.J., Gilli, A.,Wacker, L., Sturm, M., Bernasconi, S.M., Smittenberg, R.H.: Long-stored soil carbon released by prehistoric land use: evidence from compound-specific radiocarbon analysis on Soppensee lake sediments. Quat. Sci. Rev. 144, 123-131, 2016. - Lane, C.S., Horn, S.P., Taylor, Z.P., Kerr, M.: Correlation of bulk sedimentary and compound-specific d13C values indicates minimal pre-aging of n-alkanes in a small tropical watershed. Quat. Sci. Rev. 145, 238-242, 2016. - Uchikawa, J., Popp, B.N., Schoonmaker, J.E., Zu, L.: Direct application of compound-specific radiocarbon analysis of leaf waxes to establish lacustrine sediment chronology. J. Paleolimn 39, 43-60, 2008

RE.: we enhanced the focus to potential pre-aging by adding another sub-section to 5.1. where these issues are discussed. Additional to the three suggested references we also added Kusch et al., 2010 and Eglinton et al., 1997.

The document would generally benefit from improved conciseness and clarity, this includes the abstract that is much too long for such a short paper. It is my opinion that the study is worthy of publication in Climate of the Past, but considerable effort will be required to improve the conciseness of the presentation and to focus the paper more effectively on the most significant conclusions (lack of diagnostic capability based on widely-applied chain-length metrics and rapid response of alkane proxies to vegetation change). Too much of the manuscript focuses on paleoenvironmental/paleoclimate interpretations that have already been made based on pollen-based interpretations in prior studies when the truly novel aspects of this manuscript seem to be the rapid response of the alkane record to change relative to other studies that highlight 'pre-aging'.

RE.: we agree that the coeval response of of n-alkane based proxies compared to pollen is – beside source evaluation of leaf wax compounds – the most important aspect of the manuscript. Our discussion about this finding is in our view sufficient and has been further enhanced by adding another subchapter about potential lag-times to section 5.1.. We also consider that the new organic geochemical data provide an

integrative proxy data set, which deserve a separate discussion in a paleoecological context – especially for the YD-termination and early Holocene which was not yet discussed in detail through the palynological data. We think that two pages for this chapter 5.3 is not too extensive (compared to almost six pages for the above mentioned issues).

General editing that should be applied to the entirety of the text: 1. The use of conjunctive adverbs (however, thus, nevertheless, etc.) are excessive throughout the text. The paper would be much more concise if sentences were restructured to omit the conjunctive adverbs altogether. 2. The authors are flipping between active and passive voice throughout the document. (Lots of examples, but see lines 28-30 on page 7 for one example: switch from 'have been reconstructed' to 'our observations'. 3. Throughout, commas should be placed before 'which' and 'but'. 4. Compound-specific needs to be hyphenated when used as an adjective. 5. The word 'this' is used too frequently in sentence strings where it is often difficult to decipher what precisely 'this' is referring to (e.g. page 2, line 28; 6. Shouldn't results and methods be in the past tense? 7. Throughout, need to be careful when designating species vs. genera. For example, page 9 Betula spp. would be a genus, not a species.

RE.: we thank the reviewer for suggestion to improve writing of the manuscript. For revision we shortened the abstract and changed passages throughout the whole text with taking special attention to the general suggestions #1-7 listed above.

Specific comments:

1. Page 5, line 9: grammar 'hexane as solvent'

RE.: this was corrected

2. Page 5, line 20: Delta V Plus should be spaced

RE.: this was corrected

3. Page 5, line 26: should this be a permille sign?

RE.: yes, it should be permille. This was corrected

4. Page 7, line 10: should be 'concentrations'

RE.: this was corrected

5. Page 7, line 14: great example of a sentence that could be much more concise: : :Source allocation of n-alkane homologues can also be derived from the carbon isotope composition.

RE.: this was corrected

6. Page 7, line 26: semicolon should be period.

RE.: this was corrected

7. Page 8, line 6-7: low concentrations of aquatic macrophyte pollen specifically? Needs to be specified.

RE.: yes, this refers to aquatic pollen. This was inserted.

8. Page 8, line 7: remove 'also'

RE.: this was corrected

9. Page 8, line 17: just say 'are likely of terrestrial origin'.

RE.: this sentence was shortened

10. Page 8, line 25: remove 'mainly'

RE.: this was corrected

11. Page 9, line 5: no need for a colon here.

RE.: this was corrected

12. Page 9, line 13: recommend different word choice for 'mainly expanding'

RE.: this was changed

13. Page 9, lines 14-16: no need for 'in the: : :' openings to both sentences

RE.: this was corrected

14. Page 10, line 16: need to specify what exactly is being correlated; also change from 'each other' to 'one another'.

RE.: this was changed to ". . . concentrations of n-alkane homologues strongly correlate with one other"

15. Page 10, no need to capitalize 'Mid-chain'

RE.: this was corrected

16. Page 11, line 20: specify which data: : :avoid 'as discussed'.

RE.: this was changed to " Similar to pollen vs. n-alkane concentrations, autocorrelations need to be considered when interpreting n-alkane based proxies"

17. Page 12, line 3: awkward phrasing 'has been discussed to be probably equivalent'

RE.: this sentence was altered

Figure 2: I am not sure that overlaying all of the homologues in the bottom panel is effective: : :there are too many lines making it difficult to decipher the trends of any one individual homologue in the plot.

RE.: despite there is some overlay within the YD, we think the plot can be deciphered

Figures 4 and 5: latin names need to be italicized.

RE.: this was corrected

---

## Author Comment (AC2) · 15 Jun 2018

The manuscript "Leaf wax n-alkane distributions record ecological changes during the Younger Dryas at Trzechowskie paleolake (Northern Poland) without temporal delay" by Aichner et al. uses high-resolution records of n-alkane distribution and pollen, and lower resolution d13C analyses, during the last deglacial period to address whether n-alkanes reflect plant community changes. This is a well-written paper describing an impressively detailed dataset, and the data support the conclusions. This paper is a step towards developing quantitative paleohydrological reconstructions using com-

pound specific hydrogen isotopes, and as such is an important contribution to the literature. An assumption that is widely made regarding leaf waxes is that mid-chain n-alkanes reflect aquatic plant productivity and long-chain n-alkanes reflect terrestrial plant productivity, with longest chain lengths potentially being produced by grasses and herbs. These assumptions are mainly based on analyses of modern plants. Very little work directly compares n-alkane distributions with plant ecosystem composition as reflected by different proxies in sediment archives, so this study is a nice test of that assumption. In addition, if one can use n-alkane chain lengths to determine or control for plant community changes when interpreting leaf wax isotope measurements, then the analyses are simpler and less expensive than analyzing pollen or macrofossils in addition to n-alkanes. The authors use appropriate methods, which could be clarified with a couple minor text additions, detailed below. These methods and results support their two main conclusions: 1. Changing measures of chain length distributions may be influenced by multiple different factors, and should therefore be examined carefully, in terms of which individual n-alkane chain lengths are actually changing, and 2. "ACL and ratios of n-alkanes are suitable integrative proxies to track major and abrupt vegetation changes in a local setting."

RE.: we thank the reviewer for the constructive suggestions to further improve our manuscript. Responses to detailed comments are given below.

Substantive comments:

1. Changes in pollen and n-alkanes in the same sediment samples are indisputable. The authors should be a bit clearer about age uncertainties when comparing their record to proxy records from other archives. For example: p 13 line 5: the authors mention a lag here, but do not mention a lag for previous intervals. The age model uncertainty varies throughout the record, but at all times in this record, the uncertainty is enough to influence timing & interpretations/comparisons with other records. p 12 line 5: The discussion of the timing of the longer ACL at 13.2 to 13.0 ka: age uncertainty at this time period is _250 years or so (based on visual inspection of the age model in

Supp Fig. 1). The age model uncertainty should be acknowledged in the discussion, as the age model uncertainties in each of the records means that the events can't be assumed to be synchronous. p 12 line 9: Similarly, the conclusion about the timing of the YD onset 170 years after GS-1 onset: the age uncertainty at 12.6 ka in this record appears to be about _150 years, so the lag could mainly be due to age model uncertainty. This should be acknowledged in the text. If the authors have evidence to suggest that the lag at this site is real, this would be a good place to present that.

RE.: uncertainties within age-models are a critical point when comparing proxy data among different records. We critically discussed this already in the initial submission for the YD-termination. Based on the comment of the reviewer we also included a sentence about the 13.2-13.0 events, which was measured in the non-varved section of the Allerød and hence is characterized by higher age-uncertainty. Concerning the YD-onset we can rely on a highly precise age-control, due to the nature of the sediment (annual laminations) from varve counting and thephrochronological anchor points, like the LST) in this core section. This approach enables the identification of leads and lags on timescales of several decades (at least in the laminated section of the core). We emphasize this stronger now at the relevant passage of the text.

2. Carbon isotopes: Isotopic measurements can be influenced by instrument drift through time. In addition, isotopic measurements can be strongly influenced by linearity (i.e., the size of the individual peaks being measured) (Kornfeld et al., 2012). The linearity effect is stronger for smaller peaks. Because the authors report d13C data on n-alkanes of widely varying concentrations in the same sample, these peaks are likely subject to strong changes in linearity, especially the smallest peaks (C23, which differs the most from the other n-alkane chain lengths in its d13C value). How do the authors account for drift and linearity effects in their d13C analyses? Do they run standards at varying concentrations and correct for the linearity effect? This is important to state, as these effects can dramatically impact isotope values. If the authors do not control for the effects of drift and linearity, they should also state that, as that limits

the degree to which the values can be interpreted.

RE.: Linearity effects of the IRMS were assessed by running standards at different concentrations. We monitor stability of this effect by running a Schimmelmann B-mix, consisting of compounds with different concentrations, on a regular basis within the measurement sequence. Based on these data, we principally exclude peaks with an intensity of < 1000 mV for evaluation and samples were concentrated in order to achieve this minimum peak intensity for the lowest abundance compounds. We added the respective information to section 3.3.

3. I have a question about each conclusion that would be helpful to clarify in the text: Conclusion 1: As far as I can understand from these results, it seems as if taking this type of data the next step and making a quantitative interpretation of compound-specific hydrogen isotope ratios still requires an independent record of plant ecology (i.e., pollen or macrofossils). Simply knowing, for example, that ACL increased because the long chain lengths increased doesn't allow for an interpretation of the ecological shift, as this shift could have been caused by one of several different plants in an ecosystem. Is that correct? If so, it could be helpful to clarify that independent ecosystem reconstructions will still be required for future quantitative isotope interpretations. If not correct, then the opposite can be clarified in the text. Conclusion 2: This dataset is a really nice illustration of the similar timing of pollen and n-alkane concentrations. These data do not address whether ecological changes lag climate changes, correct? Perhaps the authors could point that out in the conclusion, as a reader may be inclined to interpret the data as such.

RE.: to conclusion 1: we think that our data show that n-alkane proxies can be a useful tool to reconstruct major ecological shifts in a local setting. While this could well help for interpretation of dD-data, multiproxy approaches i.e. including palynological data will always remain useful, but maybe could be conducted in lower resolution. Palynological approaches can deliver more specific information (i.e. which plant type appeared or disappeared) while n-alkane ratios rather reflect compositional changes

(i.e. if certain ratios change, vegetation has changed, but we can't pinpoint this change to individual species). We think that this opinion is sufficiently represented e.g in the abstract "Our results demonstrate, that a combination of palynological and n-alkane data can be used to infer the major sedimentary leaf wax sources and constrain leaf wax transport times from the plant source to the sedimentary sink ". To conclusion 2: we think it is clearly described that we discuss potential time lags between different types of proxies. Lags between climatic triggers and proxy response are a different objective which – as we think – does not need to be treated in the conclusion. We employ proxies mainly sensitive to vegetation change (which of course is likely driven by a climatic change). Compound-specific hydrogen isotope analysis for example could then provide independent information about climatic changes in a next step.

Minor comments:

Fig 1 a: what are the blue lines? I'm confused by the legend: it seems like the white lines with triangles are the kettle holes, but they're listed in the key as subglacial channels? I don't see the symbol for a kettle hole anywhere on the map. Can the symbols be clarified?

RE.: we updated Fig. 1 by mainly removing unnecessary information in order to clarify the issues as listed by the reviewer

p 13 line 1: I think the authors mean 11,540 years ago

RE.: this was corrected

p 13 line 24: ACL is increasing through this Early Holocene interval, not decreasing, as stated in the text

RE.:this was corrected
* * *